# In Vivo and In Vitro Expression of iC1, a Methylation-Controlled J Protein (MCJ) in Bovine Liver, and Response to In Vitro Bovine Fatty Liver Disease Model

**DOI:** 10.3390/ani13061101

**Published:** 2023-03-20

**Authors:** Shanti Choudhary, Michelle LaCasse, Ratan Kumar Choudhary, Mercedes Rincon, Donald C. Beitz, Eric D. Testroet

**Affiliations:** 1Department of Animal and Veterinary Sciences, University of Vermont, Burlington, VT 05446, USA; 2Department of Immunology & Microbiology, University of Colorado Anschutz School of Medicine, Aurora, CO 80045, USA; 3Department of Animal Science, Iowa State University, Ames, IA 50011, USA

**Keywords:** bovine, fatty liver disease, mitochondrial complex I inhibitor, expression

## Abstract

**Simple Summary:**

Fatty liver disease (FLD) is a common metabolic disorder of high-milk-yielding cows. Though FLD has been a common disorder of the cow for a considerable period, its molecular mechanism and novel strategy to prevent or mitigate economic losses are less explored. In this study, we, for the first time, showed expression of mitochondrial complex 1 inhibitor (iC1) in bovine liver in situ and established an in vitro model for studying FLD. The role of iC1 in cows should be analogous to its roles in mice and humans and may find application in the etiology and pathology of bovine FLD.

**Abstract:**

Mitochondrial complex I inhibitor (iC1) is a methylation-controlled J protein (MCJ) that decreases cellular respiration by inhibiting oxidative phosphorylation. Recent rodent studies showed that loss or inhibition of iC1 was associated with preventing lipid accumulation. A common metabolic disorder of dairy cattle is a fatty liver disease (FLD), which often occurs during the periparturient period. In humans and rodents, iC1 is expressed in the liver and acts as a mitochondrial “brake”. However, iC1 expression in bovine liver and its possible role in FLD development have not yet been characterized. We hypothesized that iC1 is expressed in the bovine liver and that the expression of iC1 is correlated with FLD in periparturient dairy cattle. To test this hypothesis, we collected bovine liver tissue samples from an abattoir and isolated primary hepatic cells immediately following harvest. Utilizing an in vitro model of bovine FLD developed in our laboratory, we cultured primary hepatic cells in low-glucose DMEM supplemented with 10% FBS. The basal media was made to induce lipid accumulation and cytotoxicity in the primary liver cells with three treatments. To the basal media (control) we added 0.4 mM palmitate (treatment 1) or 20 ng/mL TNFα (treatment 2), or both 0.4 mM palmitate and 20 ng/mL TNFα (treatment 3). Consistent with our hypothesis, we present the novel characterization of iC1 expression in primary bovine liver cells cultured with or without the addition of lipotoxic factors made to emulate bovine FLD. We demonstrate both in situ and in vitro expression of iC1 in bovine liver and mRNA expression in hepatic cells and in the precipitates of conditioned media. The results of RT-qPCR, IHC, and western blot all demonstrated the expression of iC1 in bovine liver. In addition, we isolated precipitates of conditioned media further demonstrated iC1 expression by RT-qPCR. The transcript of iC1 tended to be more concentrated (4-fold; *p* > 0.05) in TNFα-treated conditioned media when compared with the control. Taken together, we present the novel finding that iC1 transcript and protein are expressed in liver tissue from dairy cattle, primary hepatic cells isolated from that liver tissue, and, finally, in the conditioned media derived from those cells. These novel findings and the prior findings on the role of iC1 in rodents and humans indicate that further investigation of the role of iC1 in the etiology and pathology of FLD in periparturient dairy cows is warranted.

## 1. Introduction

Fatty liver disease (FLD) is a metabolic disorder common in high-producing dairy cattle, particularly around the time of calving (periparturient period). After parturition, a number of metabolic periparturient dairy cattle undergo a dramatic metabolic “transition” around the time of parturition. Initiation of lactation is metabolically demanding for any mammal requiring dramatic metabolic remodeling. One such alteration easily observed in dairy cattle is the increased lipolysis of adipose tissue reserves. During the periparturient period, blood non-esterified fatty acids are elevated, and the fatty acid composition of the liver is altered. The extent of mobilization of energy reserves from adipose tissue can be readily observed during this period as the profile of fatty acids in the liver begins to closely match the fatty acid profile of the adipose tissue [1]. The alterations in liver composition are even evident at the gross pathological level during FLD as disease severity progresses because the liver will present with a yellowish color and deteriorated structural integrity.

The physiology of the periparturient dairy cow is well characterized and it is generally accepted that fatty liver incidence can be decreased through improved management practices like specific diet to prevent milk fever. However, while management has certainly improved the physiology of periparturient cows, about 30% to 40% of dairy cow still develop mild to moderate fatty liver disease [2], indicating management alone cannot alleviate this problem. It is also worth noting that fatty liver disease alone is comparatively not very detrimental to the agricultural industry.

About half of all dairy cows, particularly when poorly managed, and especially when over-conditioned, experience FLD around parturition [3,4]. Indeed, appropriate management and nutrition programs are critical strategies to decrease periparturient disease [5]. However, even with ideal management and nutrition, cows still develop FLD, indicating that other factors are involved [6]. Some cattle develop severe FLD while transitioning from non-lactating to lactation, indicating that the body condition score alone is not an adequate predictor of transition cow disease. Other cows in the same herd often suffer the development of secondary co-morbidities associated with disruption of hepatic metabolic function (e.g., retained placenta, left-displaced abomasum, ketosis, and mastitis) [7].

After parturition, lipids are mobilized from adipose tissue and synthesized from some of the products of rumen fermentation [8]. The periparturient period is akin to a starvation state because the physiological limitations of intake prevent the cow from consuming adequate nutrients to support lactation [9]. Primarily caloric intake and protein intake are insufficient to support maximal milk synthesis [10]. The adipose-derived lipids are shuttled to the liver where they are either utilized for ATP production through mitochondria typically repackaged as VLDL or transported further to tissues and ultimately oxidized to produce ATP [11]. However, the cow does not synthesize fatty acids de novo in the liver, and thus the capacity for export to utilize excess lipid or TAG by lipolysis via oxidation or by mobilization to export to other organs results in accumulation of excess lipids in the liver and leads to FLD. Free fatty acids contribute to the liver-TAG pool. Palmitate, a saturated free fatty acid (FFA), is the most prevalent fatty acid in the liver of overfed dairy cows within the first week of calving [12]. Saturated fatty acids induce hepatocyte lipoapoptosis in the c-Jun N-terminal kinase (JNK)-dependent fashion [13]. Upregulation of lipid metabolism and inflammatory cytokines genes (*Mcp1, Tgfb1*, and *Timp1*) in murine hepatocyte cultured in the presence of TNFA demonstrated that TNFA is pivotal for the progression of non-alcoholic fatty liver disease [14]. Therefore, the effects of palmitate on hepatic cells are of particular interest towards the development of an in vitro model of bovine fatty liver disease. Because of the development of FLD, the productivity and reproductive performance of cows decreased in addition to the development of other metabolic diseases such as ketosis, milk fever, and other comorbidities.

iC1 is a transmembrane protein of the inner mitochondrial membrane that interacts with complex 1 of electron transport chain (ETC). Various tissues including heart, kidney, lung, and liver of mice and human tissue including some immune cells showed localization of iC1 [15,16]. Absence of iC1 expression has been shown to increase complex 1 activity and hence the mitochondrial respiration. Murine studies have shown that iC1 deficiency prevents accumulation of fat in the liver [15]. However, in bovine, expression of iC1 in the liver or any other organs has not been examined. We thought that the liver accumulates fat and the formation of mitochondrial respiratory complexes through mitochondrial negative regulators could lead to decreased production of ATP and increased generation of ROS in the liver. Therefore, examining expression the of iC1 in bovine liver for possible associations with the prevention of over accumulation of fat should be conducted.

Our objective was to demonstrate iC1 expression in bovine liver tissue and bovine primary hepatic cells. We hypothesized that iC1 is expressed in bovine liver and the expression of iC1 is correlated negatively with accumulation of lipid in hepatocytes. In humans and in mice, suppression of iC1 expression results in the upregulation of mitochondrial oxidative respiration, and is associated with decreased lipid accumulation, therefore making it an attractive target for prevention of FLD in dairy cattle. To test our hypothesis, we utilized our in vitro model of hepatic lipidosis [17,18]. In vitro models for the liver have been used in humans to study non-alcoholic fatty liver disease [19]. Our model uses primary bovine hepatic cells isolated from abattoir-derived tissue collected immediately after slaughter. Using exogenous TNFA and palmitic acid supplementation, we evaluated the relationship of iC1 expression to hepatic lipid accumulation. Beyond economic considerations, ultimately, our approach for collection and culture of primary bovine hepatic cells also has potential to be used for the study of elite dairy animals using tissue collected through puncture biopsy.

## 2. Materials and Methods

Liver samples were randomly collected from four culled Holstein dairy cows (n = 4) on the same day immediately following slaughter. Since the culled cows were meant to be slaughtered, about 10 g of liver tissue was collected from the dead animals from the left lobe of the liver, placed in sterile PBS (kept on ice) containing antibiotic and antimycotic, transported to our laboratory, and subdivided for primary culture. Unique to our research, we utilized primary hepatic cells isolated from adult dairy cows rather than from bull calves, which have traditionally been chosen because of the relative lack of value when compared with female dairy cattle. Liver samples of (0.5–1.0 g size) were collected for RNA isolation and histochemical studies.

### 2.1. Isolation of Primary Hepatocytes of Bovine

Bovine hepatic cells were isolated as described in [14]. Briefly, bovine liver tissues were collected from an abattoir in sterile PBS (premixed with 1% antibiotic and antimycotic solution). After washing (3 times with PBS)**,** samples were chopped using sterile scalpels and micro scissors in the digestion medium (DMEM supplemented with 3.7 g/mL sodium bicarbonate + 200 U/mL of collagenase type II) and transferred to 50 mL conical tubes sealed with parafilm to avoid any spills followed by incubation for 30 min in a heated incubator/shaker (MAXQ4000, Thermo Fisher Scientific, Waltham, MA, USA; 37 °C at 90 rpm). After digestion, the tissue slurry was passed through 1 mL pipette (front tip cut) several times to facilitate digestion. To stop further digestion, equal amounts of complete medium (DMEM + 10% FBS + 1% penicillin/streptomycin) were added to conical tubes containing digested tissue. The tissue was allowed to settle approximately for 2 min and the supernatant was transferred to new 50 mL conical tubes, using 100 µm syringe filters. The tissue pellet was then washed by mechanical disruption in 10 mL serum-free DMEM, centrifuged at 500× *g* for 5 min, and the supernatant was discarded. The cell pellet was washed (3×) using 5 mL serum-free DMEM. Supernatant was removed without disturbing cell pellets and resuspended in complete culture medium followed by seeding the pellet onto 100 mm gelatin-coated cell culture plates and T25 flasks in culture media and incubation for 48 h at 37 °C, 5 % CO_2_, 95% O_2_. After 48 h of incubation, a fresh medium was added and left for 4 days to allow cells to grow. Medium was changed every 48 h after 4 days of incubation until reaching to 70–80% sub-confluency. Sub-confluent cells were split using trypsin Trypsin/EDTA (Sigma Aldrich, St. Louis, MO, USA). Passaged hepatocyte cells (approximately 2.4 × 10^5^ cells/mL) were expanded by seeding onto 48-well, 24-well, and 96-well culture plates and experiments were conducted.

### 2.2. Preparation of TNFA and Palmitic Acid Solutions

Tumor necrosis factor alpha (TNFA, 100 µg Acro Biosystems, Newark, DE, USA) was reconstituted in 500 µL sterile water per the manufacturer’s instructions to form a 200 µg/mL stock solution, aliquoted, and stored at −70 °C until use with the final concentration being 20 ng/mL. The concentration of TNFA was chosen based upon a previously published report to induce inflammatory responses in hepatocytes [20].

A 300 mM sodium palmitate (Pal) stock solution was prepared by saponifying palmitic acid (Sigma Aldrich, St. Louis, MO, USA) in equimolar NaOH at 65 °C. Saponification was necessary because palmitic acid is inherently very insoluble in aqueous solutions. This stock solution was then further diluted with ultrapure water to 30 mM, which was heated at 80 °C for at least 30 min and added directly to 10% FBS/DMEM with the final concentration being 0.4 mM. The palmita then was allowed to complex with the albumin present in the FBS component of the culture medium for at least 60 min at room temperature prior to the treatments. The palmitate concentration was chosen to elicit maximal intracellular lipid loading without inducing massive cytotoxicity [21]. To mimic fatty liver disease condition, isolated liver cells were grown in the presence of palmitate, TNFA, and a combination of Pal + TNFA.

### 2.3. Oil Red O Staining of Hepatic cells

Following a 24 h treatment, hepatic cells were washed twice with PBS and stained with Oil Red O stain (Thermo Fisher Scientific, Waltham, MA, USA). Briefly, hepatic cells were fixed in 4% cold paraformaldehyde for 10 min. After a brief washing with PBS (×1 min), fixed monolayers of hepatic cells were rinsed with 60% isopropanol. Ready to use 0.36% Oil Red O stain in 60% isopropanol (EMD Millipore Corp. Burlington, MA, USA) was liberally applied onto the cells for 15 min and rinsed briefly with 60% isopropanol to remove excess. Finally, cells were lightly stained with hematoxylin (#3536-16; Fisher Scientific, Pittsburgh, PA, USA) for 20 s and rinsed with distilled water (2× 3 min). Cells were mounted in aqueous VECTASHIELD^®^ Antifade Mounting Media (Vector Laboratories, Newark, CA, USA) and visualized under an epifluorescence microscope (Evos M5000 Imaging System; Invitrogen, Carlsbad, CA, USA).

### 2.4. Gene Expression

Total RNA was isolated from (1) bovine liver tissue (which was stored in RNAlater (Thermo Fisher Scientific, Waltham, MA USA) at −80 °C), (2) primary hepatocytes, and (3) conditioned media collected from hepatocytes culture after the treatment of primary hepatic cells using PureLink RNA Mini Kit (Thermo Fisher Scientific, Waltham, MA, USA). The cDNA was synthesized using iScript™ Reverse Transcription Supermix for RT-qPCR kit (BioRad Lab., Hercules, CA, USA) followed by qPCR of iC1 gene-specific primers using SYBR Green chemistry on a CFX 96 thermocycler (BioRad Lab. Hercules, CA, USA). Gene-specific primer pairs were designed in house and were used for the detection of transcripts of iC1. For the detection of iC1 by real-time qPCR (CFX96; BioRad Lab, Hercules, CA, USA), we used two pairs of primer sets (IDT, Newark, NJ, USA), i.e., primer 1 forward 5′- TTGGGTGTAAGCCCATCTGC -3′; primer 1 reverse, 5′- CCTTGAGCGTCCATCAGTGT -3′; and primer 2 forward, 5′-GGGTATTGCAGCTCTTGGCT-3′ and primer 2 reverse, 5′- CTGGCAGATGGGCTTACACC -3′. The specificity of qPCR product amplification was visualized by running the qPCR product through 1% agarose gel electrophoresis and running melt curve analysis after PCR amplification.

### 2.5. Protein Isolation and Western Blot

Total protein from bovine liver tissue was isolated using RIPA lysis buffer (Thermo-Fisher Scientific, Waltham, MA, USA). About 100 mg of frozen bovine liver tissue was lysed in 600 μL of cold RIPA lysis buffer with a protease inhibitor cocktail (Sigma Aldrich, St. Louis, MO, USA) and homogenized for 2 min using a steel probe-based homogenizer. Whole-tissue lysates were then incubated at 4 °C for 2 h and briefly vortexed before centrifugation at 13,400× *g* for 20 min at 4 °C. Protein concentration of lysates was determined by the Bradford colorimetric assay (BioRad Lab, Hercules, CA, USA) with bovine serum albumin (BSA) as a protein standard. Isolated protein was aliquoted and stored at −80 °C until further use.

Wes analysis was performed on a Wes system (Biotechne, Minneapolis, MN, USA; product number 004–600) using a 12–230 kDa Wes Separation Module (ProteinSimple SM-W006), anti-Rabbit detection module (ProteinSimple DM-001), and primary antibody against iC1 (Anti-rabbit). Briefly, lysates of bovine liver were diluted to an appropriate concentration (200 μg/mL) in sample buffer (10x Sample Buffer 2, a component of separation module) and then mixed with fluorescent master mix and heated at 95 °C for 5 min and immediately placed onto ice until analysis. The following components were loaded into a pre-filled microplate—samples, antibody diluent, primary antibodies (diluted 1: 50 in antibody diluent), HRP-conjugated secondary antibodies, and chemiluminescent substrate. The plate was centrifuged (×1000 *g* for 5 min) prior to the run. Capillary electrophoresis was run with the default settings for protein signal detection.

### 2.6. Immunostaining

A sub-sample of liver tissue (0.5 × 0.5 × 0.5 cm^3^) was dissected immediately at the abattoir. Following collection, tissues were fixed in 10% neutral buffered formalin overnight. The following day, formalin was replaced with 70% ethanol and kept at 4 °C until further processing at pathology units of University of Vermont, Burlington, VT USA. A 5 µm thick section of bovine liver tissue was placed on positively charged slides and immunostained by using a previously described procedure [22]. Briefly, antigen was retrieved using Tris-EDTA based buffer (pH 9.0). Antibody incubations conditions were an overnight incubation at 4 °C with the primary antibody and an overnight incubation at room temperature with the HRP-conjugated antibody. Finally, slides were dehydrated and mounted prior to viewing at 40× objective.

Primary cells underwent their respective treatments for 24 h and then were washed with PBS (3× 3 min) and subsequently fixed in ice-cold 4% paraformaldehyde (40 min). After washing cells with PBS, protein-blocking buffer 2.5% horse serum (20 min) was applied. Cells were incubated with primary antibodies against alpha smooth muscle actin (A2547; Sigma-Millipore; 1:5000 dilution) and E-cadherin (3195S; Cell Signaling; 1:500 dilution) overnight for immunostaining. Alpha smooth muscle actin is a marker of hepatic stellate cells [23,24]. Secondary antibodies, i.e., VectaFluro’s anti-mouse DyLight 594 or anti-rabbit DyLight 488 (Vector Lab Inc., Newark, CA, USA), were applied (30 min) after being washed with PBS. Next tissue was counterstained with mounting media (ProLong™ Diamond Antifade Mountant) containing DAPI (Thermo Fisher Scientific, Waltham, MA, USA). Images were captured immediately after immunostaining in a semi-dark room using an Evos M5000 imaging system (Thermo Fisher Scientific, Waltham, MA, USA).

## 3. Results and Discussion

### 3.1. Primary Hepatocyte Culture of Bovine

We obtained viable primary hepatocytes from adult bovine liver. Tissue was collected from the slaughterhouse and was confirmed to have typical epithelial cell morphology. The histological features of hepatocytes include hexagonal cell boundaries with 1–2 round nuclei/cell (Figure 1a), whereas the stellate cells were fibroblast-like cells with elongated nuclei. Immunostaining of cells with alpha-smooth muscle actin (SMA) showed elongated fibrous-like cells (green arrows, Figure 1b,c) usually located in the periphery of hepatocyte cell aggregates and seen at the periphery culture plates. In a 3D co-culture experiment, culture of hepatocytes with stellate cells provided a better functionality of hepatocytes in comparison with 2D and 3D hepatocyte monolayer cultures alone [25]. In mixed culture of bovine hepatic cells, we successfully induced fatty liver disease-like conditions by the treatment of palmitate and TNFA. In comparison with the control culture, cells with no palmitate or TNFA showed very few lipid droplets inside the hepatic cells (Figure 1c). Hepatic stellate cells preferentially store lipid droplets, especially multiple-retinoid-containing lipid droplets [26]. A combination of palmitate and TNFA heavily induced accumulation of lipid droplets inside the hepatocyte and stellate cells (green arrow; Figure 1d). One limitation of this study was that the parity and stage of lactation of the culled animals were not recorded. Therefore, the physiology of liver cells and the difference in response to the treatment on four cows could not be compared. Establishing primary hepatic cultures of heifers, primiparous, multiparous, and then early-to-late-lactating parous animals to evaluate treatment effects is warranted.

### 3.2. Gene Expression of iC1 in Bovine Liver

To examine gene expression of iC1 in bovine liver and bovine primary hepatic cell culture, we quantified mRNA levels by RT-qPCR. With the two sets of primer pairs, designed from the different regions of the gene, we examined amplification of iC1 in liver tissue. Results showed that primer 1 and primer 2 produced two specific and desired PCR products of size 166 bp and 189 bp, respectively. Agarose gel runs of amplicon (Figure 2a) and melt cure analysis showed two specific peaks (Figure 2b). The expression of iC1 in the bovine liver tissue showed successful gene amplification (Figure 2c). In bovine hepatic cell culture, the conditioned media collected after 24 h of treatment showed differential expression in all the treatment cultures in comparison with the control culture (Figure 2d). The expression level of iC1 tended to be higher (4-fold; *p* > 0.05) in TNFA-treated conditioned media vs. the control. Transcripts of iC1 were also detected in bovine primary hepatic cells by real-time qPCR (data not shown).

### 3.3. Protein Expression of iC1 in Bovine Liver

We next examined the protein expression of iC1 in bovine liver using histology and IHC. The histology of bovine liver showed a normal cyto-structure. A representative picture of bovine liver showed pink eosinophilic cytoplasm of hepatocytes with intracytoplasmic non-staining vacuoles (Figure 3a). The liver is primarily composed of hepatocytes. Hepatocytes are the polygonal epithelial cells filled with eosinophilic granular cytoplasm, generally binucleated, and contain 1–2 prominent nucleoli. Hepatocytes are richer in mitochondria, the location of the iC1 protein. Hepatocytes may contain intracytoplasmic, non-staining vacuoles which are the cytoplasmic lipid. Metabolic diseased condition or the physiologic nutritional status of animals may be reflected by the increased amount of intracellular lipids in the hepatic cells.

Immunolocalization of iC1 in bovine mammary tissue was seen in the cytoplasm and around the nucleus of hepatic cells and its distribution in the liver parenchyma was homogenous (Figure 3b). Localization iC1 protein in the cytoplasm and especially around the periphery of nuclei was noticed in primary hepatic cells (Figure 3c). The staining patterns were similar, being located around the nuclei, but tissue appeared to have more homogenous and cytoplasmic patterns than the cultured cells. A possible reason for this difference could be due to the more functional and active state of hepatic cells in vivo- rather than the in vitro-grown hepatic cells. Differences in the staining intensity of iC1 among the various treatment groups to induce FLD in primary hepatic cells indicate variations in iC1 expression and a possible association with FLD. The omission of primary antibodies in immunostaining showed no positive signals inside the cells (Figure 3d).

Standardization of protein using the Wes method requires the optimization of antibody and protein concentrations to detect desired the protein signals [27]. Herein, we showed the application of a capillary-based immunoassay (Wes method) to detect iC1 protein in the bovine liver. Bovine-specific anti-rabbit iC1 primary antibody was selected to detect specific protein signals. In addition to this, we also validated our Wes method using published iC1 anti-mouse antibody and showed the specific protein signal of the desired size (not shown). To set up the Wes method as a new method for protein quantification, we optimized the three-antibody dilution and used an optimal dilution (1:50) to achieve a maximum signal at a fixed amount of loaded protein (Figure 3e). Likewise, we also used various amounts of loaded protein with optimum antibody dilutions and found the best signal with minimum noise. This enabled us to show the linearity of the iC1 signal. After standardization of primary antibody dilution, protein concentration, bovine liver cell lysate (sample), HeLa cell lysate (positive sample), water (negative control 1), and lysate with omission of primary antibody (negative control 2) were run and showed signals in bovine liver and positive samples and no signals in negative controls (Figure 3f).

## 4. Conclusions

In conclusion, we detected the expression of mitochondrial complex I inhibitor (iC1) in bovine liver tissue and isolated primary hepatic cells. We detected iC1 at the gene and protein levels using RT-qPCR and the Wes method, respectively. In future, the role of iC1 in the development of bovine fatty liver disease would be worthy to investigate for its possible therapeutic promises. Mitigating FLD in dairy cows and selecting FLD-resistant animals would eventually contribute to developing a sustainable dairy sector.

## Figures and Tables

**Figure 1 animals-13-01101-f001:**
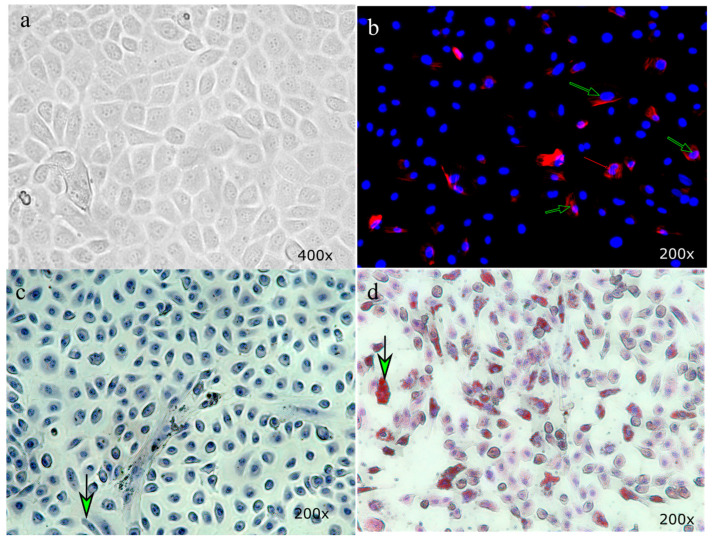
Treatment of bovine hepatic cells with TNFA and palmitate induced lipid accumulation. A representative image showing transillumination images of the majority of the cuboidal-shaped hepatocytes with a mixture of a few other cell types such as stellate cells (**a**). Stellate cells are fibroblast-like cells which are alpha-smooth muscle actin-positive (**b**). Induction of fatty liver disease by palmitate and TNFA, evaluated by lipid droplet staining using Oil Red O stain, showed no to very little storage of lipid droplets inside the cells (**c**) in comparison with heavy accumulation of lipid droplets in treatment of cells with a cocktail of palmitate and TNFA (**d**). Arrows in panel (**b**–**d**) indicate stellate cells.

**Figure 2 animals-13-01101-f002:**
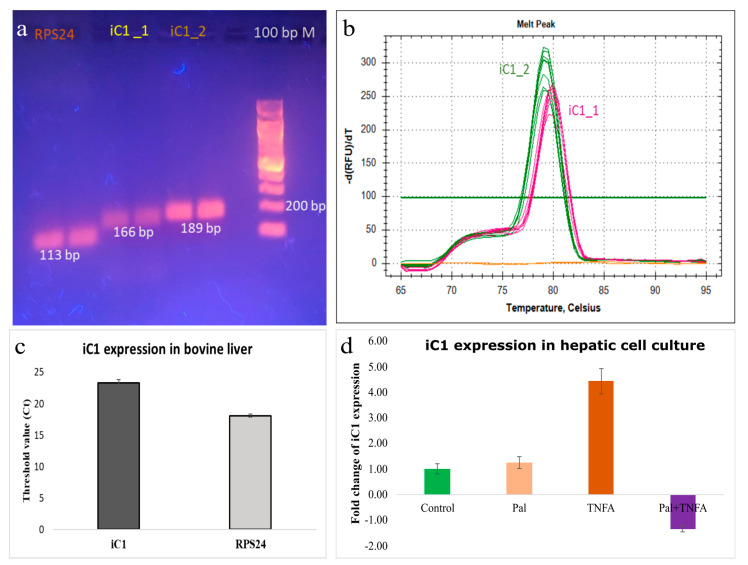
Gene expression of iC1 in bovine liver. Transcript of iC1 was tested using two pairs of primers (iC1_1 and iC1_2), resulting in successful amplification of a single amplicon (166 bp and 189 bp, respectively) shown by agarose gel electrophoresis (**a**). Melt curve analysis carried out after the end of qPCR amplification showed a single melt peak, suggesting a single PCR product amplified by each primer pair (**b**). Gene expression of iC1 in bovine liver tissue (n = 4), expressed in terms of threshold (Ct) value, showing successful amplification of iC1 and the endogenous reference gene, *RPS24* (**c**). Results of iC1 expression in conditioned medium are in fold change normalized to the *RPS24* gene with the control culture as a calibrator (**d**). Conditioned media of bovine hepatic cells collected 24 h post treatment with palmitate (Pal), tissue necrosis factor alpha (TNFA), and cocktail of Pal + TNFA. No-reverse transcriptase (NRT) and no-template control (NTC) were the negative controls of RT-qPCR amplification.

**Figure 3 animals-13-01101-f003:**
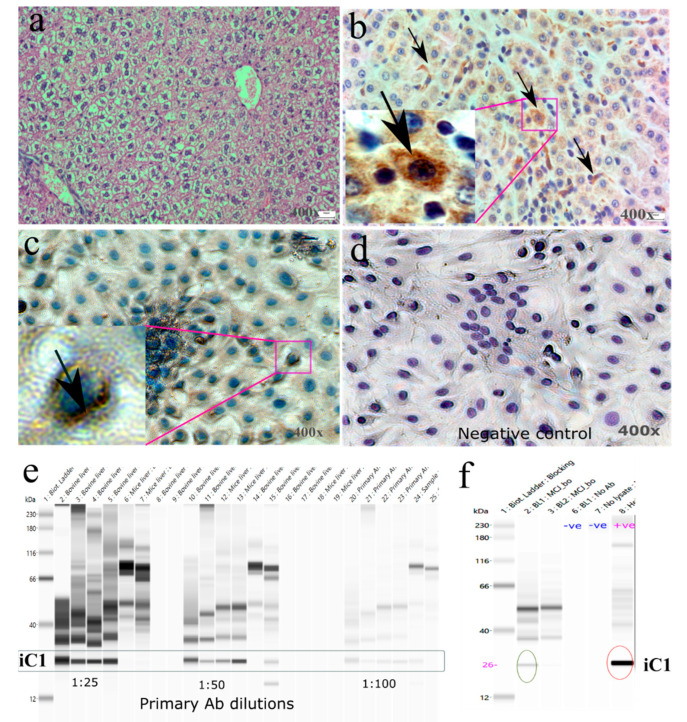
Protein expression of iC1 in bovine livers. Hematoxylin and eosin staining of bovine livers showing the morphology of healthy bovine livers obtained from slaughter (**a**). Immunolocalization of iC1 was observed in the cytoplasm of hepatic cells (arrows and magnified images of cellular localization of iC1 are shown) in liver tissue (**b**) and primary hepatic culture (**c**). Negative runs of iC1 immunostaining performed by the omission of primary antibody showed no positive brown staining (**d**). Standardization of anti-rabbit primary antibody dilution of iC1 in capillary Western blot (ProteinSimple of the Wes model) showed a specific gel-like band of iC1 protein in bovine liver tissue lysates, showing a decreased intensity of the protein band as the antibody was diluted further (**e**). After standardization of primary antibody dilution, the iC1 band is shown in tissue lysate (**f**). The omission of primary antibody and no-cell lysate were the two negative controls. The lysate of HeLa cell was used as a positive control of Wes runs. Protein band of iC1 in bovine liver and HeLa cells are encircled.

## Data Availability

The data presented in this study are available in the manuscript.

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
