# Peer review of "In Vivo and In Vitro Expression of iC1, a Methylation-Controlled J Protein (MCJ) in Bovine Liver, and Response to In Vitro Bovine Fatty Liver Disease Model"

_animals, 2023, doi:10.3390/ani13061101_

Round 1
Reviewer 1 Report
The paper fits well with the scope of the journal. However, in my opinion, it has some shortcomings in this current form. In particular, my concerns are:
- manuscript need to be consistent with the journal reference style (https://www.mdpi.com/journal/animals/instructions)
- reference section need to be checked
Specific comments:LL61-63: please include citation, for example you can read and cite 10.1080/1828051X.2021.1990804
L71: citation?
LL73-74: include citation (e.g., 10.1080/1828051X.2022.2032850)
LL83-84: Paper Malhi et al 2006 is not consistent with the reference section
Reference Chavez-Tapia et al., 2011 is not mentioned in the text
Author Response
The paper fits well with the scope of the journal. However, in my opinion, it has some shortcomings in this current form. In particular, my concerns are:
- manuscript need to be consistent with the journal reference style (https://www.mdpi.com/journal/animals/instructions)
AU: Thank you for the comment. The references have been fixed.
- reference section need to be checked
Specific comments:
AU: added
LL61-63: please include citation, for example you can read and cite 10.1080/1828051X.2021.1990804
AU: Reference added
L71: citation? –
AU: Reference included
LL73-74: include citation (e.g., 10.1080/1828051X.2022.2032850)
AU: Reference included
LL83-84: Paper Malhi et al 2006 is not consistent with the reference section
AU: Needful done.
Reference Chavez-Tapia et al., 2011 is not mentioned in the text
AU: Reference of Chavez-Tapia has been added (in line no L105)

Reviewer 2 Report
Same as below
Reviewer 3 Report
In the present manuscript, the authors investigate the expression of Mitochondrial complex I inhibitor (iC1) in bovine liver and its possible role in fatty liver disease (FLD) development. This would be of potential importance, but there are several deficiencies that need to be addressed or clarified for further improvement of the manuscript.
1. In Figure 1c and d, the authors evaluated lipid droplet by Oil Red O stain. More inflammation parameters should be investigated in order to establish the in vitro bovine fatty liver disease model.
2. In Figure 1, Figure 3c and d, please add a scale bar for all panels.
3. In Figure 1b, alpha-smooth muscle actin-positive is typically used to characterize smooth muscle cells. Please explain why it is used to characterize stellate cells.
4. In Figure 2d, please add an error bar accordingly.
5. Closer attention needs to be paid to small details. For example, Introduction, Line 1, "Non-communicable diseases greatly increased than Communicable diseases". Communicable should not be capitalized. Please revise the whole manuscript carefully.
Author Response
Comments and Suggestions for Authors
In the present manuscript, the authors investigate the expression of Mitochondrial complex I inhibitor (iC1) in bovine liver and its possible role in fatty liver disease (FLD) development. This would be of potential importance, but there are several deficiencies that need to be addressed or clarified for further improvement of the manuscript.
- In Figure 1c and d, the authors evaluated lipid droplet by Oil Red O stain. More inflammation parameters should be investigated in order to establish the in vitro bovine fatty liver disease model.
AU: Accumulation of lipid droplet is good in vitro model to study pathophysiological changes seen in hepatic steatosis (Nutrients 2023, 15(1), 40; https://doi.org/10.3390/nu15010040).
- In Figure 1, Figure 3c and d, please add a scale bar for all panels.
AU: Magnification of the images are duly mentioned over the picture.
- In Figure 1b, alpha-smooth muscle actin-positive is typically used to characterize smooth muscle cells. Please explain why it is used to characterize stellate cells.
AU: Alpha smooth muscle actin is a definite marker of hepatic stem cells has been added with appropriate references (line 227-228; Ref # 21,22). Thank you.
- In Figure 2d, please add an error bar accordingly.
AU: Needful done.
- Closer attention needs to be paid to small details. For example, Introduction, Line 1, "Non-communicable diseases greatly increased than Communicable diseases". Communicable should not be capitalized. Please revise the whole manuscript carefully.
AU: Manuscript has been revised thoroughly for such mistakes. Thank you.

Round 2
Reviewer 1 Report
The authors have responded to all the comments raised in the previous review. From my point of view, the manuscript can be published.
Author Response
Dear Reviewer,
Thank you for your time in critically evaluating our manuscript.
Ratan
Reviewer 2 Report
Authors have addressed most of my comments.
There is a comment regarding the sampling collection process (Line 118) that hasn't been addressed. It's important for authors to describe the features of the animals, such as parity and stage of lactation, that were sampled since these important factors can affect liver cells physiology, altering morphology and structure of these cells, especially when referring to fatty liver disease. If the authors don't posses that information, this should be mentioned as a pitfall of the study.
Author Response
Dear Reviewer,
Thank you for pointing out your concern. Specifically,
There is a comment regarding the sampling collection process (Line 118) that hasn't been addressed.
AU: We have addressed that concern and added information that samples were collected from a culled and dead animal (Line no 123-125).
It's important for authors to describe the features of the animals, such as parity and stage of lactation, that were sampled since these important factors can affect liver cells physiology, altering morphology and structure of these cells, especially when referring to fatty liver disease. If the authors don't posses that information, this should be mentioned as a pitfall of the study.
AU: We agree with your concern. To address this, we have included a statement in lines 266-270. Manuscript is attached (PDF file to verify my claims).
We believe that these changes made in the revised manuscript would be acceptable to you.
